# The Adiponectin Receptor Agonist, ALY688: A Promising Therapeutic for Fibrosis in the Dystrophic Muscle

**DOI:** 10.3390/cells12162101

**Published:** 2023-08-19

**Authors:** Nicolas Dubuisson, Romain Versele, Maria A. Davis-López de Carrizosa, Camille M. Selvais, Laurence Noel, Chloé Planchon, Peter Y. K. Van den Bergh, Sonia M. Brichard, Michel Abou-Samra

**Affiliations:** 1Endocrinology, Diabetes and Nutrition Unit, Institute of Experimental and Clinical Research (IREC), Medical Sector, Université Catholique de Louvain (UCLouvain), Avenue Hippocrate 55, 1200 Brussels, Belgium; nicolas.j.dubuisson@uclouvain.be (N.D.); romain.versele@uclouvain.be (R.V.); mayadavis@us.es (M.A.D.-L.d.C.); camille.selvais@uclouvain.be (C.M.S.); laurence.noel@uclouvain.be (L.N.); chloe.planchon@student.uclouvain.be (C.P.); sonia.brichard@uclouvain.be (S.M.B.); 2Neuromuscular Reference Center, Department of Neurology, Cliniques Universitaires Saint-Luc, Avenue Hippocrate 10, 1200 Brussels, Belgium; peter.vandenbergh@uclouvain.be; 3Departamento de Fisiología, Facultad de Biología, Universidad de Sevilla, 41012 Seville, Spain

**Keywords:** duchenne muscular dystrophy, adiponectin, ALY688, skeletal muscle, AMPK, inflammation, regeneration, fibrosis, myonecrosis, necroptosis

## Abstract

Duchenne muscular dystrophy (DMD) is one of the most devastating myopathies, where severe inflammation exacerbates disease progression. Previously, we demonstrated that adiponectin (ApN), a hormone with powerful pleiotropic effects, can efficiently improve the dystrophic phenotype. However, its practical therapeutic application is limited. In this study, we investigated ALY688, a small peptide ApN receptor agonist, as a potential novel treatment for DMD. Four-week-old mdx mice were subcutaneously treated for two months with ALY688 and then compared to untreated mdx and wild-type mice. In vivo and ex vivo tests were performed to assess muscle function and pathophysiology. Additionally, in vitro tests were conducted on human DMD myotubes. Our results showed that ALY688 significantly improved the physical performance of mice and exerted potent anti-inflammatory, anti-oxidative and anti-fibrotic actions on the dystrophic muscle. Additionally, ALY688 hampered myonecrosis, partly mediated by necroptosis, and enhanced the myogenic program. Some of these effects were also recapitulated in human DMD myotubes. ALY688’s protective and beneficial properties were mainly mediated by the AMPK-PGC-1α axis, which led to suppression of NF-κβ and TGF-β. Our results demonstrate that an ApN mimic may be a promising and effective therapeutic prospect for a better management of DMD.

## 1. Introduction

Duchenne muscular dystrophy (DMD) is one of the most prevalent inherited myopathies, affecting 1/3500 boys [1]. DMD remains a rapidly progressive and lethal disorder, where patients are typically wheelchair bound by 8–14 years of age and die from cardiac or respiratory failure during their third decade [2]. This myopathy is caused by mutations in the gene encoding dystrophin, a protein that provides structural stability and integrity to the myofibre membrane [1]. Mutations in dystrophin result in membrane damage, allowing severe inflammation/oxidative stress and necrosis [1,3]. Ongoing cycles of muscle necrosis and repair cause exhaustion of satellite cells and impaired regeneration capacity. Eventually, muscle fibres are gradually replaced by fibrosis, leading to muscle wasting and weakness [1,4]. Although dystrophin mutation is the primary cause of DMD, it is the secondary processes involving persistent inflammation, impaired regeneration and prolific fibrosis that actually worsen the course of the disease [3,4].

Adiponectin (ApN) is a hormone abundantly secreted by adipocytes. It promotes insulin-sensitizing, fat-burning and anti-inflammatory/oxidative actions, thereby effectively counteracting several metabolic disorders, including type 2 diabetes, obesity and cardiovascular disease [5]. ApN exerts its pleiotropic effects in a variety of cell types through binding to its main receptors, AdipoR1 and AdipoR2 [6]. AdipoR1, which has a high affinity for both the full-length and the cleaved globular fragment of ApN (gApN), is mainly expressed in skeletal muscle, whereas AdipoR2 is predominantly expressed in the liver [7]. As skeletal muscle is a main target tissue of ApN, we tested this hormone in mdx mice, a mouse model of DMD. When we crossed mdx mice with transgenic mice overexpressing ApN, this hormone delayed disease progression by reducing muscle inflammation/injury and improving force/myogenesis [8]. Conversely, when we crossed mdx mice with ApN-knockout mice, the resulting mdx mice displayed a reverse picture, which was corrected by muscular electro-transfer of the *ApN* gene. Thus, ApN may be a powerful protector of the skeletal muscle [9]. However, there are some limitations in using ApN directly as a therapeutic agent. Its complex quaternary structure and rapid turnover impede its production in sustained amounts [5]. As a result, the search for novel compounds with AdipoRs agonist activity, which may further be easily produced, has been developing over the last decade.

ALY688 (formerly known as ADP355) is a decapeptide AdipoRs agonist which is derived from the active site of human gApN and acts preferentially through AdipoR1. This first-in-class small peptide agonist was discovered as an inhibitor of cancer cell growth that was more potent than gApN [10,11]. Later on, several other beneficial metabolic effects were observed both in vitro and in vivo. More specifically, when administered to mice, ALY688 reduced inflammation and fibrosis after toxic liver or heart injury [12,13,14]; it also attenuated insulin resistance induced by a high-fat/high-sucrose diet [15] and inhibited atherosclerosis in apoE-deficient mice [16]. As yet, its potential anti-inflammatory, pro-myogenic and anti-fibrotic properties have not been tested in skeletal muscle and a fortiori not in the dystrophic muscle.

The aim of this study was to explore whether ALY688 may play a beneficial role in DMD. To this end, ALY688 was administered subcutaneously (sc) for a period of 2 months, starting early in mdx mice since muscle degenerative-regenerative cycles begin as soon as 3–4 weeks of age [17]. We first examined whether treated mice showed reduced skeletal muscle inflammation, oxidative stress and fibrosis as well as enhanced muscular function. Then, we uncovered the potential mechanisms of action underlying the effects of ALY688. Finally, we also evaluated some of these effects in primary cultures of human myotubes originating from healthy subjects and DMD patients.

## 2. Materials and Methods

### 2.1. Animals

*C57BL/10ScSn-DmdmdxJ* mdx mice (murine model of DMD) and *C57BL/10ScSnJ* mice [used as wild-type (WT) controls] were purchased from Jackson Laboratory (Bar Harbor, ME, USA). The cohort was divided into four groups of male mice. Each group comprised 10 mice. The first group was composed of WT mice, the second one of untreated mdx mice (mdx), while the third and fourth groups were composed of mdx mice treated with a slow release (SR) form of ALY688 at either 3 mg/kg/day or 15 mg/kg/day (mdx-T3 or mdx-T15). Compound dosage and treatment duration were based on Allysta pharmaceuticals’ preliminary experiments. ALY688 (SR formulation) was administered daily by subcutaneous (sc) injection for 2 months. Regular mdx mice were injected with saline. Animals were maintained under a standard laboratory chow and housed at a constant temperature (22 °C) with a fixed 12 h light to 12 h dark cycle (lights on from 7 a.m. to 7 p.m.). Twelve-week-old mice were sacrificed between 09.00 and 11.00 h. Pairs of muscles [*Quadriceps* (Q), *Gastrocnemius* (G), and *Tibialis anterior* (TA)] were weighed, frozen in liquid nitrogen and stored at −80 °C for subsequent analyses. These are mixed muscles with similar fibre type composition [18,19].

### 2.2. In Vivo Studies of Muscle Function

At week 11 (i.e., 1 week prior to sacrifice), mice were submitted to three widely used and reliable functional tests [20,21,22]. These tests were performed successively, one test per day over a 5-day period.

Wire test. Mice were suspended by their limbs from a wire and the time until they completely released their grasp and fell was recorded. Mice that reached a set limit of 600 s were allowed to stop the experiment, while others were directly retested, up to three times, and their maximum hanging time was recorded. The holding impulse (body mass × hang time), used to oppose the gravitational force, was then calculated [22]. 

Grip test. Limb strength was recorded using a grid connected to a sensor (Panlab-Bioseb, Vitrolles, France). Mice were gently laid on the top of a grid so that their front paws could grip the grid. Mice were then pulled back steadily until the grip was released down the complete length of the grid. Each test was repeated three times at an interval of 15 min. Results are presented as the mean of the three values of force recorded, related to body weight [21].

Treadmill exhaustion test. After two consecutive days of acclimation to the moving belt (Panlab-Bioseb), mice were submitted to a treadmill exhaustion test with an upward inclination of 5° and increasing speed rate over 4 steps. The mice started by running 10 min at a pace of 20 cm/s (step 1), then 5 min at 25 cm/s (step 2), followed by 5 min at 30 cm/s (step 3) and finally 5 min at a maximum speed of 35 cm/s (step 4). Exhaustion was defined as the inability of the animal to run on the treadmill for 5 s despite laying on top of the shock grid and receiving repeated aversive stimuli [23,24,25]. The distance covered in meters (m) during the test was recorded either after exhaustion or at the end of the test. If the mouse was able to complete all steps, the total test duration was 25 min for a maximal distance covered of 390 m.

### 2.3. Bright-Field Histochemistry

G muscles were fixed in 10% formalin for 24 h and embedded in paraffin. Immunohistochemistry was carried out as previously described [20,21]. Briefly, 5 μm sections were processed using antibodies directed against interleukin 1-beta (IL-1β), tumour necrosis factor alpha (TNFα), peroxiredoxin 3 (PRDX3), 4-hydroxy-2-nonenal (HNE) and cluster of differentiation 68 (CD68) (Appendix A). Before immunostaining, sections were submitted to heat-mediated antigen retrieval using a microwave oven and Tris-citrate buffer (pH 6.5). Binding of antibodies was detected by applying for 30 min at room temperature a secondary antibody, which was a biotinylated goat anti-rabbit IgG (H + L). For each marker, all slides were treated simultaneously for immunohistochemistry analysis and diaminobenzidine revelation (DAB, Thermo Fisher Scientific, Waltham, MA, USA) and then analysed together. Whole muscle sections were scanned, and then the percentage of areas stained with DAB was quantified using QuPath (opensource, https://qupath.github.io, accessed on 26 July 2022, Belfast, UK). TA sections were also stained with haematoxylin and eosin to evaluate myonecrosis. Quantification of myonecrosis is shown as the proportion (%) of whole muscle section area occupied by fibres with fragmented sarcoplasm and inflammatory cells [26]. In addition, Q sections (described below) were stained with Picrosirius red (Abcam, Cambridge, UK) to evaluate muscle fibrosis. Fibrotic tissue was scored setting a colour balance threshold and data obtained were expressed as the percentage of total section area.

### 2.4. Immunofluorescence

Q muscles were embedded in optimum cutting temperature medium (OCT; VWR International, Dublin, Ireland) and frozen in liquid nitrogen chilled isopentane (VWR International). A total of 10 µm transversal cryosections were fixed with 4% paraformaldehyde and blocked with 10% goat serum. Antibodies directed against dystrophin and laminin were used (Appendix A). Secondary antibodies were AF488 and AF647-conjugated goat anti-rabbit or anti-rat (Sigma-Aldrich, St-Louis, MO, USA), respectively. Finally, nuclei were stained with DAPI (Thermo Fisher Scientific). Whole slides were scanned with a fluorescence microscope (Axio Scan.Z1, Zeiss, Oberkochen, Germany). Dystrophin-positive fibres were counted and expressed as the percentage of the total number of fibres per muscle (ZEN 3.4 blue edition, Zeiss). A dystrophin-positive revertant fibre (RF) was scored when more than half of its membrane circumference expressed a green positive signal [27]. The number of clusters (with at least two adjacent RFs) and the maximal number of RFs in a cluster were counted on whole sections as well [27]. Centrally nucleated fibres (CNF) were also calculated using a personalized version of MuscleJ [28] based on the automatic detection of laminin and DAPI immunofluorescence.

### 2.5. Culture of Murine C2C12 Cell Lines

Murine C2C12 myoblasts were cultured in Dulbecco’s modified Eagle’s medium (DMEM), 10% Foetal bovine serum (FBS), 1% non-essential amino acids, 1% L-glutamine and 1% antibiotic–antimycotic (all from Thermo Fisher Scientific) at 37 °C in 5% CO_2_. Briefly, after reaching 80–90% confluence, the growth medium was substituted by a fusion medium, where 10% FBS was replaced by 2% heat-inactivated horse serum (HS; Thermo Fisher Scientific) for 6 days to induce myogenic differentiation. The differentiation medium was changed every other day. Then, myotubes were treated or not with mouse recombinant TNFα (10 ng/mL) + mouse interferon gamma (IFNγ) (10 ng/mL) (TNFα and IFNγ from PeproTech, Hamburg, Germany) and/or ALY688 (100 nM) for 24 h. The ALY688 concentration was chosen based on a previous study [15] and on our preliminary experiments. ALY688 used in all in vitro experiments was the conventional form of the compound (i.e., not the slow-release one). At the end of the experiments, myotubes were rinsed twice in cold PBS before RNA extraction.

### 2.6. Culture of Human Myotubes

Primary cultures of human skeletal muscle cells were initiated from myoblasts of DMD patients (*n* = 4; age range: 12–15 years) and healthy subjects (*n* = 3; age range: 15–17 years), which were provided by the French Telethon Myobank-AFM. Myoblasts were grown in DMEM/F-12 supplemented with 20% FBS, 1% non-essential amino acids, 1% L-glutamine, and 1% antibiotic-antimycotic (all from Thermo Fisher Scientific) at 37 °C in 5% CO_2_. After reaching a density of 80–90%, the growth medium was substituted by a fusion medium, where 20% FBS was replaced by 2% HS, and differentiation was allowed to continue for 14 days (time required to obtain mature myotubes with characteristic elongated and multinucleated morphology). Medium was changed every other day. Cells were always used at passages between 4 and 10. At day 14, myotubes were either left untreated, or treated with AdipoRon (25 µM) or ALY688 (conventional formulation) at different concentrations (from 10 pM up to 300 nM) for 24 h, while being challenged or not with TNFα (15 ng/mL) + IFNγ (15 ng/mL) (both from PeproTech) for 24 h.

In some experiments, cells were first transfected before inflammatory challenge and ALY688 (100 nM) treatment. Briefly, cells were transfected with either the On-Targetplus non-targeting pool siRNAs (negative control, NT siRNAs) or a specific On-Targetplus siRNA SMARTpool against human AdipoR1 (50 nM) (from Dharmacon, Thermo Fisher Scientific) using 7 μL of Lipofectamine RNAiMAX reagent (Thermo Fisher Scientific) for 24 h. RNA silencing was effective, ranging around 95% in all experiments. Next, the medium was renewed and cells were treated with ALY688 (100 nM) and with TNFα + IFNγ (15 ng/mL each) for an additional 24 h.

At the end of the experiments, cells were rinsed twice in cold PBS before RNA or protein extraction.

### 2.7. RNA Extraction and Real-Time Quantitative PCR

RNA was isolated from muscle tissues or cultured cells with TriPure reagent (Sigma-Aldrich). RT-qPCR primers for mouse *cyclophilin*, Collagen Type I Alpha 1 Chain (*COL1A1*), Collagen Type III Alpha 1 Chain (*COL3A1*), oestrogen receptor-related alpha (*ERRα*), Myogenic regulatory factors 4 (*Mrf4*), *Myh1* and *Myh7* are provided in Appendix A. RT-qPCR primers for human TATA box-binding protein (*TBP*), *AdipoR1*, *IL-1β*, *TNFα* and Utrophin (*UTRN*) are also indicated in Appendix A. Threshold cycles (Ct) were always measured in duplicate.

### 2.8. Protein Extraction and ELISAs

Muscle samples and cultured cells were homogenized in a lysis buffer supplemented with 1% protease/phosphatase inhibitor cocktail (both from Cell Signaling Technology, Leiden, The Netherlands) and 10 mM NaF (Sigma-Aldrich). Protein levels were quantified using the Bradford method and 10–150 μg of total protein extracts were used for ELISAs.

ELISA assays allowed us to specifically detect and quantify HNE (Abcam), Myogenin, Myh7, UTRN (all from Antibodies Online, Atlanta, GA, USA), the active and phosphorylated forms of AMP-activated protein kinase (AMPK), receptor-interacting protein (RIP) family of threonine/serine kinases, Smad2, p65 subunit of nuclear factor-kappa B (NF-κB) (all from Cell Signaling Technology) as well as TNFα, peroxisome proliferator-activated receptor γ coactivator-1α (PGC-1α) and active transforming growth factor-β (TGF-β) (all from MyBiosource, San Diego, CA, USA) (Appendix A; [20,21,29,30]). Kits were based on colorimetric methods and were carried out following manufacturer’s instructions.

### 2.9. Statistical Analysis

Results are means ± standard error of mean (SEM) for the indicated number of mice, C2C12 cells or human subjects. When the four groups of mice (WT, mdx, mdx-T3 and mdx-T15) were compared, the differences between groups were assessed by a one-way analysis of variance (ANOVA) followed by Tukey’s test. In one experiment (RFs), as WT values were undetectable, the 3 groups of dystrophic mice were compared by a one-way ANOVA followed by Dunnett’s test (treated mdx vs. untreated ones). When comparisons between two myotube conditions from a given subject were made, a two-tailed paired Student’s *t* test was used. In vitro dose-response curves were generated using nonlinear regression. The analysis provided the median inhibitory (IC_50_) or activating concentration (EC_50_). Comparisons between data points were then carried out by repeated measures ANOVA followed by a post hoc Dunnett’s test, where every mean was compared to the condition without ALY688, which was arbitrarily set up to be a very low concentration of the compound (i.e., 10^−12^ M). All statistical analyses and graph fittings were achieved with Prism 9 (GraphPad Software, Inc., San Diego, CA, USA). Differences were considered statistically significant at *p* < 0.05.

## 3. Results

### 3.1. ALY688 Enhances Force and Endurance of Mdx Mice

Two different doses of ALY688SR were injected sc to mdx mice for 2 months, starting at 4 weeks of age. Mdx littermates were separated into three groups: those which were treated with a daily dose of ALY688 at 3 or 15 mg/kg/day (mdx-T3 and mdx-T15) and those who were left untreated (mdx). These three groups were also compared with untreated wild-type (WT) control mice. ALY688 treatment did not affect the total body weight of dystrophic mice, nor the weight of the different removed muscles (Appendix A). Macroscopic evaluation of the liver and kidney after treatment was also unremarkable, thus indicating that the treatment was well tolerated.

To evaluate the effects of ALY688 on muscle function, mice were subjected to three functional tests: the wire test, the grip test and the treadmill exhaustion test. The wire test gives an indication of muscle fatigue and coordination. In this test, the time during which the mouse is suspended on a horizontal wire is measured. Mdx mice fell down much faster than WT mice, mdx-T3 mice showed intermediate laps of time (+34% vs. mdx), while mdx-T15 showed remarkable improvement (+67% vs. mdx) and reached normal values (Figure 1A).

The grip test measures the strength of limb muscles. The force developed by four limbs was decreased in mdx mice, while being rescued by ALY688 (+20% and +13% in mdx-T3 and mdx-T15, respectively) (Figure 1B).

The uphill treadmill exhaustion test provides valuable data on muscle endurance. The speed rate was gradually increased over four consecutive steps. The distance covered by mdx mice was significantly reduced, while that of treated mice was not different from controls (Figure 1C). More specifically, a higher number of treated mice were able to succeed in the last step despite the incremental speed, thereby confirming improved muscle endurance compared to regular mdx (Figure 1D).

Taken together, these data indicate enhanced force and resistance to fatigue in dystrophic muscle under ALY688 treatment.

### 3.2. ALY688 Effectively Reduces Muscle Inflammation and Oxidative Stress

We next studied inflammation and oxidative stress in dystrophic muscle and tested the hypothesis that ALY688 could slow down the progression of these events (Figure 2).

When compared to WT mice, myofibres from G muscles of mdx mice displayed a strong immunolabeling for inflammatory cytokines (IL-1β and TNFα), oxidative stress markers (a lipid peroxidation product, HNE and an antioxidant enzyme, PRDX3) and the macrophage marker, CD68. Immunolabelling for all these markers of inflammation or oxidative stress was attenuated in treated mice (Figure 2).

Quantification of DAB staining confirmed that these parameters in mdx-treated mice were either significantly decreased compared to regular mdx (IL-1β: −54% and −40%; PRDX3: −49% and −43%; CD68: −47% and −44% in mdx-T3 and mdx-T15, respectively) or less significantly different from those of WT (Figure 3A–E). In addition, TNFα and HNE protein levels measured by ELISAs were also significantly decreased (−40%) in whole Q muscle homogenates from treated vs. untreated mdx animals, thereby strengthening the anti-inflammatory and anti-oxidative effect of the compound (Figure 3F,G).

In conclusion, ALY688 reduced inflammation, oxidative stress and infiltration of CD68^+^ macrophages in treated mdx mice.

### 3.3. ALY688 Counteracts Myonecrosis in Mdx Mice

Because inflammation may subsequently lead to muscle damage and necrosis, we hypothesized that ALY688 could also reduce myonecrosis. This parameter was assessed by inflammatory infiltrates and fragmented myofibres and quantified on H&E-stained TA sections. Myonecrosis was significantly reduced in both treated groups (~−60%) compared to regular mdx (Figure 4A,B). In addition, protein levels of P-RIP, a novel marker of necroptosis in DMD [31] was also significantly decreased in treated mice (−25%; Figure 4C). There was no significant difference in the percentage of central nucleated fibres (CNF) in Q muscles from 3 groups of dystrophic mice (~60% for all 3 groups). However, this parameter is of limited use in identifying any striking reduction in recent active myonecrosis and subsequent regeneration in adult mdx mice [32].

Taken together, these results indicate that ALY688 markedly attenuates myonecrosis by inhibiting, at least in part, the necroptosis pathway in dystrophic mice.

### 3.4. ALY688 Increases the Number of Revertant Myofibres and Enhances the Myogenic Program in Mdx Mice

Because inflammation and cell damage may subsequently lead to muscle regeneration, we hypothesised that ALY688 could promote a muscle healing process. Studies were performed in Q muscles.

First, we studied the presence of revertant fibres (RFs), which serves as an index of muscle regeneration capacity [27]. RFs are dystrophin-positive myofibres expressed at low percentages both in DMD patients and mdx mice [27]. We thus detected by immunofluorescence the expression of dystrophin, which is located near the sarcolemma. There were almost no RFs in Q sections of mdx mice, while mdx-T3 and -T15 mice displayed four to five times more immuno-positive fibres, respectively (Figure 5A,B). Likewise, the number of clusters, (with at least two adjacent RFs) and the maximal number of RFs within a cluster, were or tended to be higher in treated mdx than in untreated ones (Figure 5B,C).

In addition, mRNA levels of *Mrf4*, a muscle marker of late differentiation was more than halved in mdx mice, while these levels tended to re-increase in treated animals (Figure 5E). Protein levels of Myogenin were decreased in mdx mice, as described [33] and restored by the treatment, while protein abundance of Myosin heavy chain, Myh7, a marker of slow-twitch oxidative type I fibres, found in mature muscle was overexpressed by ALY688 (Figure 5F,G). In vitro experiments in murine C2C12 myotubes cultured in an inflammatory context to mimic the dystrophic microenvironment unambiguously confirmed those data. ALY688 induced a strong restoration of both *Mrf4* and *Myh7*. In agreement with improved oxidative phenotype, gene expression of *ERRα*, a marker of mitochondrial biogenesis, was also normalized (Appendix A). *Myh1* mRNAs, a marker of fast-twitch glycolytic type II fibres was not or quasi not modified both ex vivo and in vitro (not shown).

Taken together, these results indicate that ALY688 may enhance the myogenic program in dystrophic muscle and its oxidative capacity.

### 3.5. ALY688 Attenuates Fibrosis in Mdx Mice

We next investigated whether administration of ALY688 could reduce fibrosis, a hallmark of muscular dystrophies and a major cause of muscle weakness [34]. Q mdx muscles displayed a strong staining for Picrosirius red, a marker of fibrosis labelling collagen types I and III. In contrast, the percentage of fibrotic areas was reduced by ~50–60% in mdx-treated mice (Figure 5B and Figure 6A). In addition, protein levels of TGF-β, a highly pleiotropic cytokine playing an important role in wound healing and inducing the production of an extracellular matrix, were increased in mdx mice but returned back to normal in mice treated with the peptide (Figure 6C). The active form of Smad2 (P-Smad2), an effector protein of the TGF-β pathway, was also higher in mdx mice than in WT ones, while being diminished in mdx mice treated with ALY688 (~45%) (Figure 6D). Additionally, gene expression of Collagen type I alpha 1 (*COL1A1*) and Collagen type III alpha 1 (*COL3A1*) was also quantified (Figure 6E,F). Treatment with ALY688 reduced the expression of *Col1A1* by half, while *Col3A1* was diminished by ~40%.

Taken together, these results indicate that muscle accumulation of fibrosis-related factors and subsequent fibrosis is strongly reduced by ALY688 treatment in mdx mice.

### 3.6. ALY688 Enhances Key Effectors of the AMPK Pathway in Mdx Mice

We examined, on TA muscles, whether ALY688 could, like ApN, stimulate the AMPK-PGC-1α axis. AMPK activity (P-AMPK) was actually increased by ALY688 (+55% vs other untreated groups; Figure 7A). PGC-1α protein levels were decreased in mdx mice (−25%) and rescued by ALY688 (Figure 7B). As the AMPK-PGC-1α axis is known to repress the activity of NF-κB, a master regulator of inflammation [20], we measured this parameter. As expected, NF-κB activity (P-p65 subunit) was ~3-fold higher in mdx than in WT mice but was then reduced by ~35% under ALY688 treatment (Figure 7C). Lastly, the protein levels of Utrophin (UTRN), an analogue of dystrophin and a target of PGC-1α [8,35], were quantified in dystrophic muscle. Accordingly, UTRN protein levels were slightly augmented in mdx mice, likely to compensate for the lack of dystrophin [8]. These levels were further slightly but significantly increased (~10%) in treated mice, possibly as a result of lower inflammation [36] (Figure 7D).

These results indicate that ALY688 potently activates the AMPK-PGC-1α axis in mdx mice, thereby decreasing NF-κB activity and upregulating UTRN.

### 3.7. ALY688 Recapitulates Its Beneficial Effects on Human Myotubes Challenged by Pro-Inflammatory Cytokines

Due to its convenience and cost-effectiveness, the mdx mouse remains the most widely used model for studying DMD [37]. However, since this model exhibits a milder phenotype than patients, it was important to validate our data in human myotubes. We thus examined the direct effects of ALY688 in primary cultures of DMD and healthy myotubes. In order to mimic the inflammatory microenvironment of DMD, we challenged the myotubes with an inflammatory stimulus (TNFα/IFNγ) for 24 h, while ALY688 was added at different concentrations (from 10 pM to 300 nM).

Firstly, we confirmed the reproducibility of the ALY688 anti-inflammatory and pro-UTRN effects in human tissue, via its action on AdipoR1 (Figure 8). Gene expression levels of *IL-1β*, *TNFα* and *UTRN* were quantified over a range of ALY688 concentrations and dose-response curves were generated. A significant reduction in *IL-1β* and *TNFα* was detected starting from 10 pM ALY688 (*p* < 0.01 or less; Figure 8A,B) and an increase in *UTRN* from 10 nM onwards (*p* < 0.01; Figure 8C). The maximal effect for each parameter was obtained at about 100–300 nM ALY688. The median inhibitory concentrations (IC_50_) for *IL-1 β* and *TNFα* were achieved around ~100 and ~35 pM, respectively, while the median activating concentration (EC_50_) for *UTRN* was a bit higher (Figure 8A–C). Similar effects of ALY688 treatment were observed on challenged myotubes derived from healthy subjects (Appendix A).

To establish the central role of AdipoR1 in ALY688 effects, human myotubes were transfected with either a non-targeting pool siRNAs (siNT as negative control) or a specific siRNA against human AdipoR1 and were then exposed to 100 nM ALY688 and the inflammatory cocktail. First, we verified the effectiveness of the siRNA against AdipoR1 (siAR1) by demonstrating that its presence almost completely abolished *AdipoR1* mRNA levels (Appendix A). Then, gene expression of the pro-inflammatory cytokines *IL-1β* and *TNFα* was quantified. The results showed a significant ~2.5- and ~3.5-fold increase, respectively, under the effect of siAR1 compared to the siNT control group (Figure 7D,E). *UTRN* gene expression level was slightly but significantly decreased by ~16% after *AdipoR1* inhibition compared to the control group (Figure 8F). Taken together, these results demonstrate the AdipoR1-dependent effects of ALY688.

Secondly, we tested the effects of ALY688 on the AdipoR1-AMPK signalling pathway. Dose-response curves were also established for the activities of P-AMPK, P-p65 and for protein levels of UTRN (Figure 9A–C). The results show positive correlations between the concentration of ALY688 and the activation of AMPK or UTRN protein abundance, while a negative correlation was found with P-p65. ALY688 was already effective on the three parameters from 100 pM onwards (*p* < 0.05 or less) with a maximal response around 100–300 nM. EC_50_/IC_50_ were within a nanomolar range. Once again, silencing AdipoR1 reversed all the effects of ALY688 (Figure 9D–F). Likewise, ALY688 was effective on challenged myotubes derived from healthy subjects (Appendix A).

Finally, we compared the effects of an optimal concentration of ALY688 (100 nM) with an optimal concentration of AdipoRon (25 µM) [20] on all these parameters. ALY688 was usually as effective as AdipoRon, but at much lower concentrations (100 nM vs. 25 µM) (Figure 10).

Taken together, these results show that ALY688 potently activates the AMPK signalling pathway in human DMD myotubes, leading to downregulated NF-κB activity and pro-inflammatory cytokine abundance, and upregulated UTRN levels, in an AdipoR1-dependent manner.

## 4. Discussion

Due to its interesting properties, we have been studying Adiponectin (ApN) for almost two decades, with a special focus on its beneficial and protective properties on muscle, as a main target tissue [5]. More recently, we started investigating its effects in Duchenne muscular dystrophy (DMD). We and others have shown low circulating levels of ApN in mdx mice [8,38], and in human patients [39], while ApN supplementation was able to counteract the progression of the disease in the mouse model [8,9]. Therefore, there is a rationale to therapeutically correcting the low levels of ApN in DMD patients.

However, the direct use of ApN as a therapeutic agent is very limited. Several small molecules and short peptides have been recently identified that target ApN receptors (AdipoRs) and mimic some of ApN’s effects. Today, two of these have been extensively characterised, AdipoRon and ALY688 (formerly known as ADP355) [5]. AdipoRon is an ApN receptor small-molecule agonist, first discovered by Okada-Iwabu [40], and is now being studied in a large range of preclinical disease models [5]. We have recently shown that daily administration of AdipoRon for two months was able to rescue the dystrophic phenotype by attenuating muscle inflammation and injury, while enhancing muscle regeneration and function [20]. However, even though AdipoRon has been commercially available for a decade, it has only been used for research purposes and no human clinical trials have been conducted. ALY688 is also an ApN receptor agonist. This decapeptide has been developed by Otvos et al., 2011 [10], and now exists in two different formulations. First, a conventional formulation, used either for in vitro studies or as an ophthalmic solution, which has been recently tested in a Phase 1/2a clinical study to evaluate the safety and efficacy in subjects with xerophthalmia (dry eyes) (NCT04201574). Second, a slow-release formulation of ALY688 (ALY688SR), administered subcutaneously, recently developed to target inflammatory and fibrotic diseases. Thus, ALY688 is a strong candidate for clinical evaluation in the near future as a promising therapeutic. One novel interest of this study was to investigate the full effect of ALY688 on the dystrophic skeletal muscle, and to potentially offer a new and promising therapeutic prospect for better management of DMD.

We showed that a daily sc administration over an eight-week period could protect the dystrophic muscle from excessive inflammatory responses and oxidative stress (Figure 11). Indeed, pro-inflammatory cytokine expression, oxidative stress markers and CD68^+^ macrophages infiltrates were reduced in mdx treated animals. Likewise, the extension of myonecrosis was decreased in treated mdx mice. The abundance of P-RIP protein, a necroptosis factor that contributes to myofibre death in DMD muscle [31], was also reduced, suggesting that this protein could at least in part play a role in the anti-myonecrotic effect of ALY688. Taken together, our findings are consistent with other studies showing that administration of ALY688 significantly reduced hepatic macrophage activation [13] as well as inflammation, oxidative stress and tissue damage after toxic heart injury [14]. 

ALY688 also positively modulated the muscle regeneration process (Figure 11). Indeed, mdx mice, as well as challenged C2C12 muscle cells, treated with ALY688 displayed increased expression of muscle differentiation and maturation factors. Similarly, other studies have demonstrated the tissue regenerating process, where treatment with ALY688 prevented cardiac atrophy and promoted liver regeneration [13,14]. Moreover, treated mice displayed more revertant fibres (RFs) than untreated mdx mice. RFs are sporadic dystrophin-positive myofibres observed in both DMD patients and mdx mice [41]. They arise from alternative splicing in satellite cells and their expansion reflects the activity of these precursor cells, and thus serves as an index of muscle regeneration capacity [27,42]. This result highlights for the first time the effect of an ApN mimic on the presence and expansion of RFs in DMD, thus strengthening the protective and regenerative effects of ALY688 on the dystrophic muscle.

Fibrosis is thought to be one prominent pathological mechanism in DMD, leading to impaired muscle function and ultimately death [4,34]. It is well known that the chronic muscle injury and inflammation, seen in DMD, leads to the recruitment of fibro-adipogenic progenitors (FAPs), which in turn differentiate into fibroblasts by TGF-β thereby increasing the deposition of connective tissues [43,44]. TGF-β is a major mediator of the fibrotic response and acts via its effector, the phosphorylated and active form of Smad2, which can then promote the formation of collagen type I and III. To date, DMD still awaits a prominent anti-fibrotic treatment [45]. In the present study, we highlight the potent anti-fibrotic properties of ALY688 (Figure 11), as demonstrated by normalised Picrosirius Red staining. In addition, the quantification of major profibrotic actors such as TGF-β and P-Smad2, as well as the end products such as collagen types I and III, were also either markedly reduced or normalised after ALY688 treatment. These results correlate with the effects of ALY688 observed previously on fibrotic hearts [14] and livers [12,13].

By exploring the mechanisms of action of ALY688, we found that AMPK was activated in ALY688-treated mice, as demonstrated by the increased phosphorylated form of AMPK and the subsequent increased levels of PGC-1α (Figure 11). This activation helps to rescue the dystrophic phenotype by four fundamental mechanisms. First, the AMPK-PGC-1α axis is a powerful suppressor of NF-κB signalling and of inflammation [46], which in turn improves the dystrophic muscle, as already demonstrated in other studies [3,8,47,48]. Second, PGC-1α, via mitochondrial biogenesis and function that are impaired in DMD [49], can contribute to the change of muscle fibre type towards an oxidative phenotype, which is more resistant to the absence of dystrophin [50]. A similar switch has been observed with AdipoRon [20]. Likewise, overactivation of PGC-1α by histone deacetylase inhibitors also mitigated the dystrophic phenotype by reverting the mitochondrial biogenesis deficit [49]. Third, expression and protein levels of utrophin, a dystrophin analogue, are increased by AMPK-PGC-1α. Up-regulation of utrophin may restore sarcolemmal integrity and confer morphological and functional improvements in mdx mice [35,51]. Fourth, AMPK can be a powerful suppressor of TGF-β/Smad signalling [52,53]. Conversely, specific AMPK inhibition in mice FAPs has been shown to enhance TGF-β signalling and promote fibrosis in regenerated muscles [54]. Moreover, macrophages also promote fibrosis in dystrophic muscles, while AMPK activation might reduce their TGF-β production and subsequently their pro-fibrotic effect [55]. Our findings demonstrate that ALY688 can be a potent AMPK signalling activator, suppressing NF-κB and TGF-β activities and upregulating utrophin. This in turn may lessen muscle inflammation, stress and fibrosis, protect muscle against injury and enhance muscle state and function, thus strikingly alleviating the dystrophic phenotype (Figure 11). Beside AMPK signalling, ALY688 could activate, similar to ApN, other cascades and effectors, such as Ca^2+^/calmodulin-dependent protein kinase kinase, p38 mitogen-activated protein kinase, peroxisome proliferator activated receptor alpha, ceramidase activity and others, to exert a plethora of metabolic and protective actions in several target tissues and organs [5,56,57,58].

A puzzling lack of a clear dose-dependent effect of ALY688SR was observed in vivo. We cannot exclude an incomplete or uneven distribution of the active compound. Alternatively, activation of some metabolic pathways/functions may follow a “bell-shaped” curve, with the 3 mg/kg/day dose already being optimal, while the beneficial effects could be limited at highest doses, as has been reported for some AdipoR agonists [59].

The mdx mouse remains one of the best-known animal models for DMD research due to its practicality and affordability [60]. Although mice express a non-functional dystrophin due to a point mutation in the DMD gene, they display a milder phenotype than DMD patients [61]. Therefore, it was crucial to confirm our findings using human myotubes from DMD patients. Hence, we showed that primary cultures of either healthy or DMD myotubes produced effects similar to those shown in mdx mice, which supports our earlier findings from our studies with ApN [8,62]. Indeed, the conventional solution of ALY688 used in vitro was extremely effective at counteracting inflammation in challenged myotubes, even at extremely low concentrations. The central role of the AdipoR1-AMPK-PGC-1α pathway was also confirmed in human myotubes, as the effects of ALY688 were totally dependent on the presence of AdipoR1.

We next assessed the effectiveness of ALY688 with that of AdipoRon both in vivo/ex vivo and in vitro. Firstly, when we compared the effects of ALY688 found in mdx mice to those previously described with AdipoRon [20], both ApN mimics effectively reduced muscle inflammation and improved muscle performance; AdipoRon had a slightly better pro-myogenic effect, while ALY688 displayed very potent anti-fibrotic properties. The anti-fibrotic potential of AdipoRon has not been explored in DMD yet, but some studies have reported that it could mitigate cutaneous, liver and renal fibrosis [63,64,65]. Secondly, when we directly compared the in vitro effects of ALY688 to those of AdipoRon on human DMD myotubes, both ApN mimics exhibited similar protective effects, but this was reached at much lower (200-fold) concentrations with ALY688 (optimal concentrations: 25 µM for AdipoRon [20] vs. 100 nM for ALY688). More interestingly, ALY688 was already effective on most parameters at a very low concentration of 100 pM, giving ALY688 a clear advantage over AdipoRon for the in vitro study.

Besides gene therapy, which is limited to a small set of patients [66,67], the only medication currently used to slow the course of DMD is glucocorticoids (GCs). Their anti-inflammatory and immunosuppressive characteristics are beneficial in the treatment of this disease [68]. Unfortunately, their prolonged usage is responsible for several adverse effects such as cushingoid facies, weight gain, glucose intolerance, growth retardation, vertebral fractures and muscular atrophy [68]. As a result, ALY688’s anti-inflammatory, pro-myogenic and potent anti-fibrotic effects on skeletal muscle make it a potentially attractive alternative to GCs. ALY688 also has the benefit of protecting liver and cardiac functions in mice and seems to be safe in our study and in other murine models, as well as in formal toxicology studies [12,13,14,16]. Moreover, unlike GCs, ALY688 could enhance insulin sensitivity and regulate adiposity, thereby improving metabolic condition [15,69].

## 5. Conclusions

In conclusion, our data show that ALY688, an ApN mimetic, could strongly induce AMPK signalling and exert potent protective effects on a dystrophic muscle. More specifically, ALY688 proved to be a powerful anti-fibrotic agent that could prevent or mitigate fibrosis in the skeletal muscle, thus providing a high priority treatment option for patients. ALY688 could also be highly impactful for the treatment of other muscles, including the heart, or for inflammatory and fibrotic diseases.

## Figures and Tables

**Figure 1 cells-12-02101-f001:**
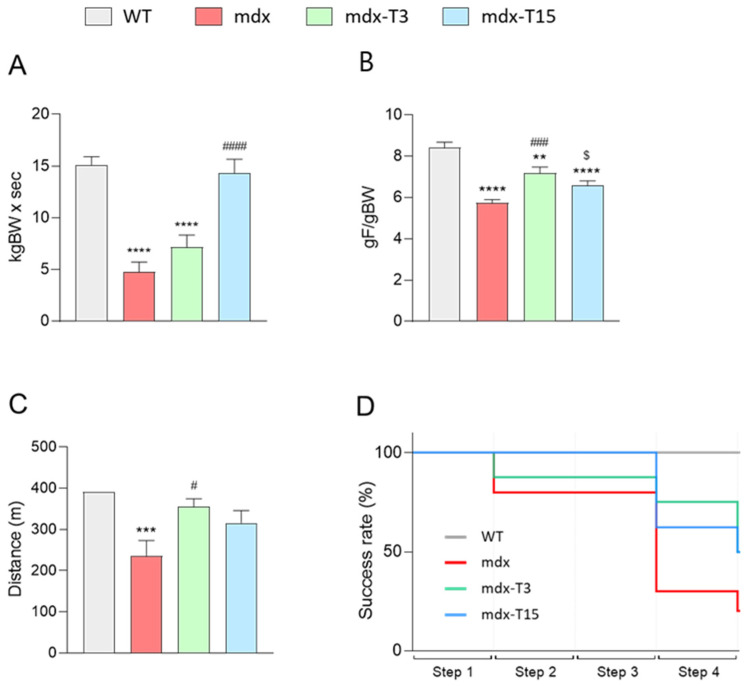
ALY688 treatment improves muscle function and endurance of mdx mice. Four groups of mice were compared at the age of 11 weeks: WT, mdx (untreated), mdx treated with ALY688 3 mg/kg (mdx-T3) and mdx treated with ALY688 15 mg/kg (mdx-T15) mice. Functional tests were carried out in vivo. (**A**) Mice were subjected to a wire test where they were suspended by their limbs and the time until they completely released the wire and fell was registered (s). This time was then normalised to body weight (kgBW × s). (**B**) Mice were lowered on a grid connected to a sensor to measure the muscle force of their four limbs; data were then expressed in gram-force relative to gram-body weight (gF/gBW). (**C**) Mice were placed on a moving belt and encouraged to run to exhaustion with an uphill inclination of 5° and a gradually increasing speed. The mice started by running 10 min at a pace of 20 cm/s (step 1), then 5 min at 25 cm/s (step 2), followed by 5 min at 30 cm/s (step 3) and finally 5 min at a maximum speed set up at 35cm/s (step 4). The distance covered in meters (m) was recorded either after exhaustion or at the end of the test. (**D**) Treadmill exhaustion test’s success rate. Data are means ± SEM; *n* = 8–10 mice for all experiments. Statistical analysis was performed using a one-way ANOVA followed by Tukey’s test. ** *p* < 0.01, *** *p* < 0.001, **** *p* < 0.0001 vs. WT mice. ^$^
*p* = 0.08, ^#^
*p* < 0.05, ^###^
*p* < 0.001, ^####^
*p* < 0.001 vs. mdx mice.

**Figure 2 cells-12-02101-f002:**
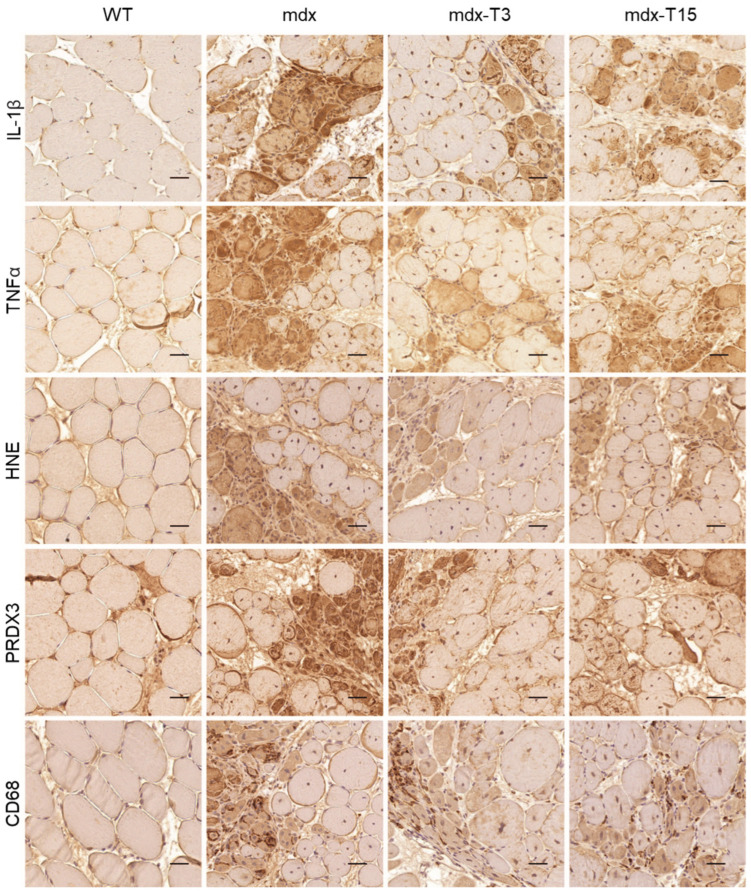
ALY688 treatment reduces muscle inflammation and oxidative stress in mdx mice. Immunohistochemistry was performed on G muscles of the 4 groups of mice. Sections were stained with specific antibodies directed against two pro-inflammatory cytokines (IL-1β and TNFα), two oxidative stress markers (HNE and PRDX3) and one pan-macrophage marker (CD68). Representative sections for 6 mice per group is shown. Scale bars = 50 μm.

**Figure 3 cells-12-02101-f003:**
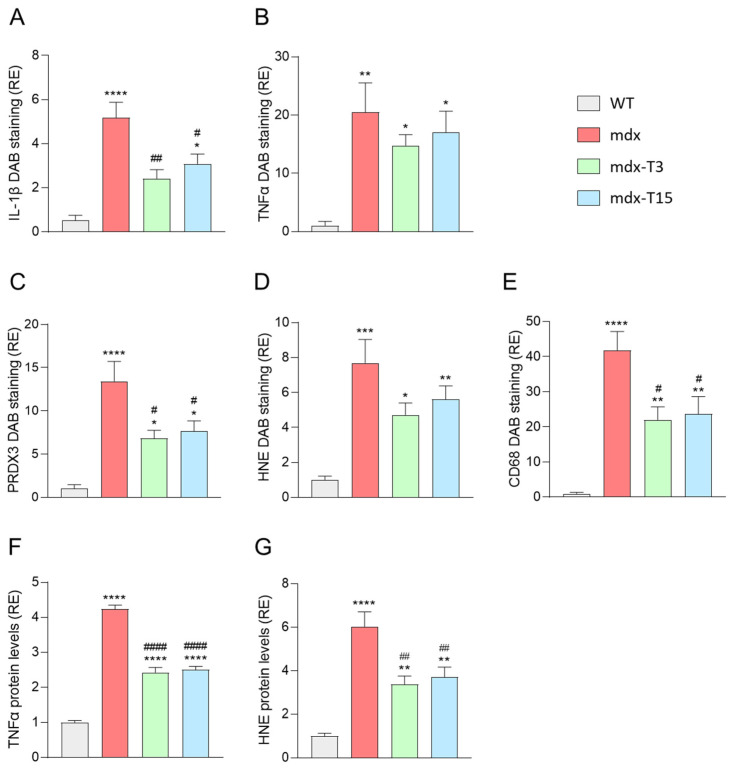
ALY688 treatment reduces muscle inflammation and oxidative stress in mdx mice. (**A**–**E**) Immunohistochemistry quantification of IL-1β, TNFα, HNE, PRDX3 and CD68 on G muscles in the 4 groups of mice. Data are calculated as the percentage area stained by DAB, then presented as relative expression compared to WT mice (RE). (**F**,**G**) ELISA quantification of TNFα and HNE protein levels in whole Q muscle homogenates; data are then presented as RE. Results are means ± SEM; *n* = 6 mice per group for all experiments. Statistical analysis was performed using a one-way ANOVA followed by Tukey’s test. * *p* < 0.05, ** *p* < 0.01, *** *p* < 0.001, **** *p* < 0.0001 vs. WT mice. ^#^
*p* < 0.05, ^##^
*p* < 0.01, ^####^
*p* < 0.0001 vs. mdx mice.

**Figure 4 cells-12-02101-f004:**
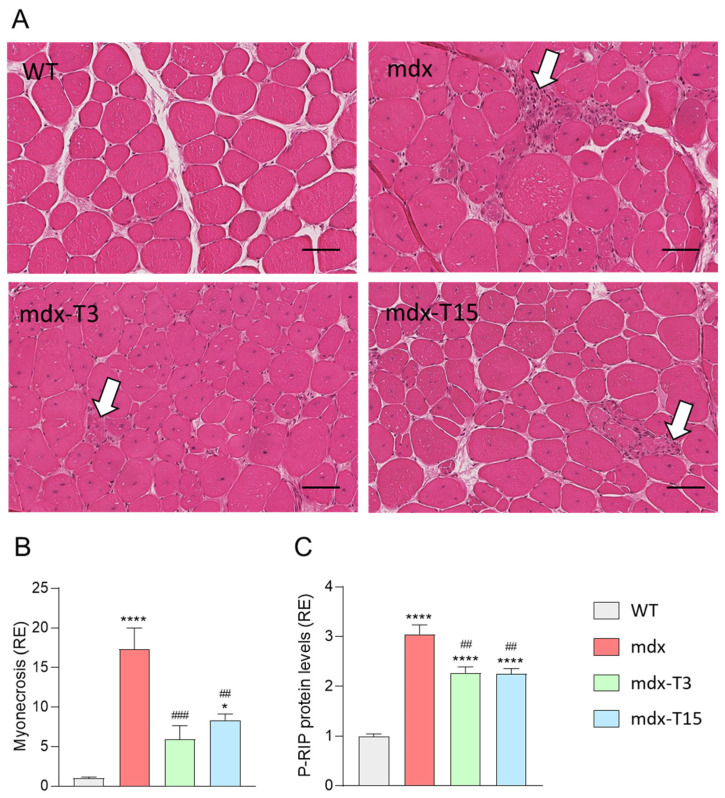
Quantification of myonecrosis in muscle sections of mdx mice. (**A**) H&E-stained transverse sections of paraffin-embedded TA muscles. Areas of myonecrosis (indicated by white arrows) encompass both muscle fibres with fragmented sarcoplasm and inflammatory cells. (**B**) Quantification is calculated as the proportion (%) of whole muscle section area occupied by myonecrosis, then presented as relative expression compared to WT mice (RE). (**C**) ELISA assays were used to quantify P-RIP, a marker of necroptosis on Q muscle; data are then presented as RE. Results are means ± SEM; *n* = 6 mice per group for all experiments. Statistical analysis was performed using a one-way ANOVA followed by Tukey’s test. * *p* < 0.05, **** *p* < 0.0001 vs. WT mice. ^##^
*p* < 0.05, ^###^
*p* < 0.01 vs. mdx mice. Scale bars: 100 µm.

**Figure 5 cells-12-02101-f005:**
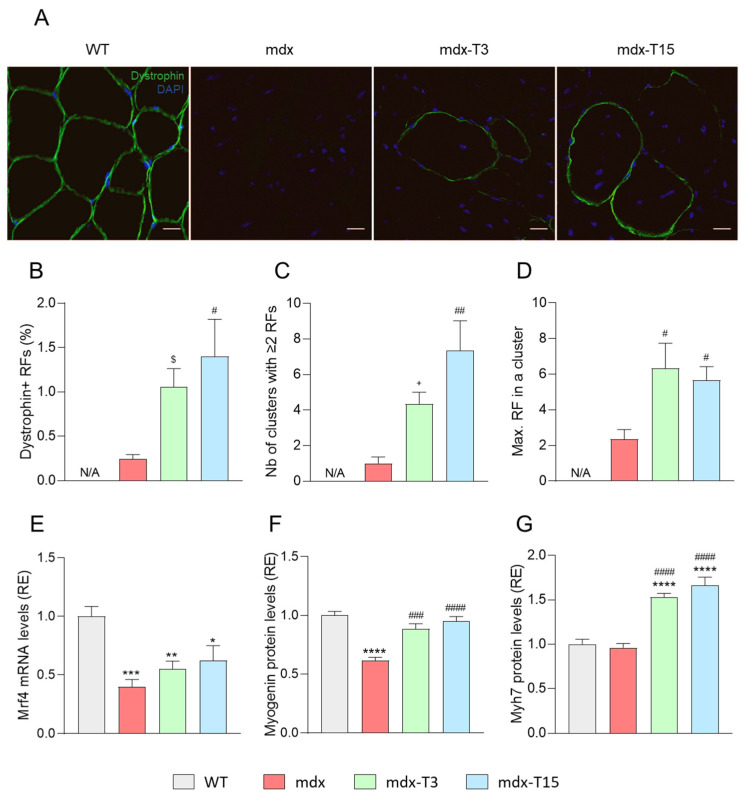
ALY688 treatment increases the number of revertant myofibres and enhances the myogenic program in mdx mice. Immunofluorescence staining, mRNA and protein abundance measurements were performed on Q muscles in the 4 groups of mice. (**A**) Sections were stained with specific antibodies directed against dystrophin (in green). Nuclei were counterstained with DAPI (blue). Scale bars = 20 μm. (**B**–**D**) Quantification of dystrophin-positive revertant fibres (RF). The number of RFs per section was counted according to the following categories: (**B**) the % of RFs, (**C**) the number of RF clusters and (**D**) the maximum number of RFs in a single cluster. (**E**) mRNA levels of *Mrf4*, a marker of late muscle differentiation was quantified, normalised to *Cyclophilin* and then presented as relative expression (RE) compared to WT values. (**F**,**G**) Protein levels of Myogenin, a differentiation marker, and Myh7, a marker of slow-twitch oxidative type I fibres, were measured by ELISA; data are then presented as RE. Results are means ± SEM *n* = 6 mice per group for all experiments. Statistical analysis was performed using a one-way ANOVA followed by Tukey’s test (or Dunnett’s test for **B**–**D**). N/A, not applicable. * *p* < 0.05, ** *p* < 0.01, *** *p* < 0.001, **** *p* < 0.0001 vs. WT mice. ^$^
*p* = 0.09, ^+^
*p* = 0.07, ^#^
*p* < 0.05, ^##^
*p* < 0.01, ^###^
*p* < 0.001, ^####^
*p* < 0.0001 vs. mdx mice.

**Figure 6 cells-12-02101-f006:**
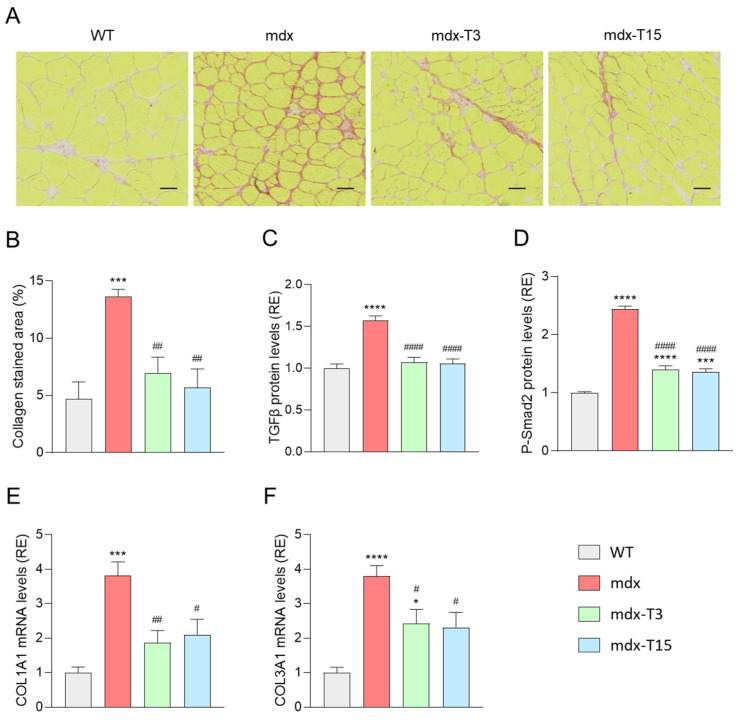
ALY688 treatment markedly reduces muscle fibrosis in mdx mice. Experiments were performed on Q muscles from the four groups of mice. (**A**) Picrosirius red staining. Scale bars = 100 μm. (**B**) Quantification of Picrosirius red (expressed as % of collagen-stained area). (**C**,**D**) ELISA assays were used to quantify TGF-β, a marker of extracellular matrix production, and phosphorylated-Smad2 (P-Smad2), an effector of the TGF-β pathway. (**E**,**F**) mRNA levels of *COL1A1* and *COL3A1*. These levels were normalised to *Cyclophilin*. Results for TGF-β, P-Smad2, *COL1A1* and *COL3A1* were presented as relative expression (RE) compared to WT values. Data are means ± SEM; *n* = 6 mice per group for all experiments. Statistical analysis was performed using a one-way ANOVA followed by Tukey’s test. * *p* < 0.05, *** *p* < 0.001, **** *p* < 0.001 vs. WT mice. ^#^
*p* < 0.05, ^##^
*p* < 0.01, ^####^
*p* < 0.0001 vs. mdx mice.

**Figure 7 cells-12-02101-f007:**
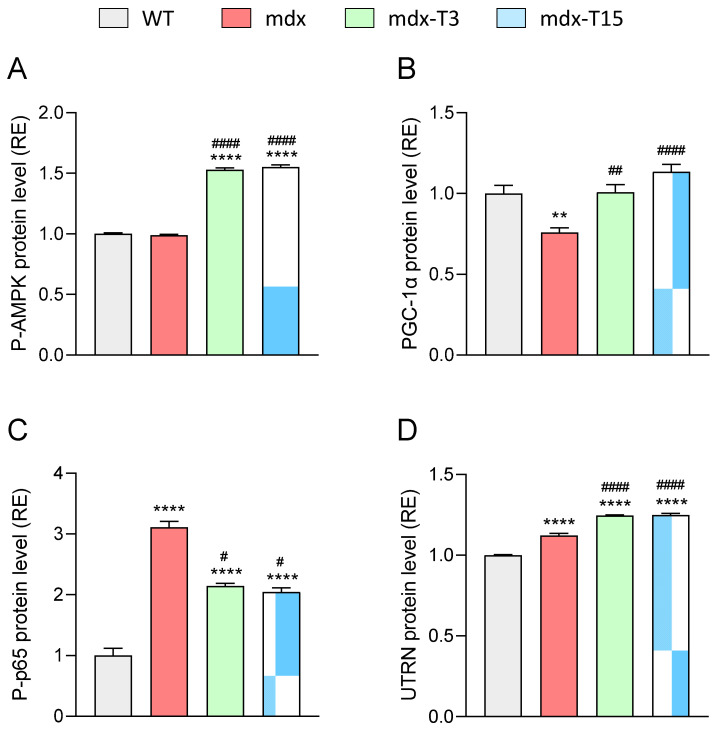
ALY688 treatment activates key effectors of the AMPK-PGC-1α axis in mdx mice. Experiments were performed on *TA* from the four groups of mice. (**A**) Levels of AMPK activity, (**B**) PGC-1α protein, (**C**) NF-κB activity (P-p65 subunit) and (**D**) UTRN protein quantified by ELISAs. Absorbance data are presented as relative expression (RE) compared with WT values. Data are means ± SEM; *n* = 6 mice per group for all experiments. Statistical analysis was performed using a one-way ANOVA followed by Tukey’s test. ** *p* < 0.01, **** *p* < 0.0001 vs. WT mice. ^#^
*p* < 0.05, ^##^
*p* < 0.01, ^####^
*p* < 0.0001 vs. mdx mice.

**Figure 8 cells-12-02101-f008:**
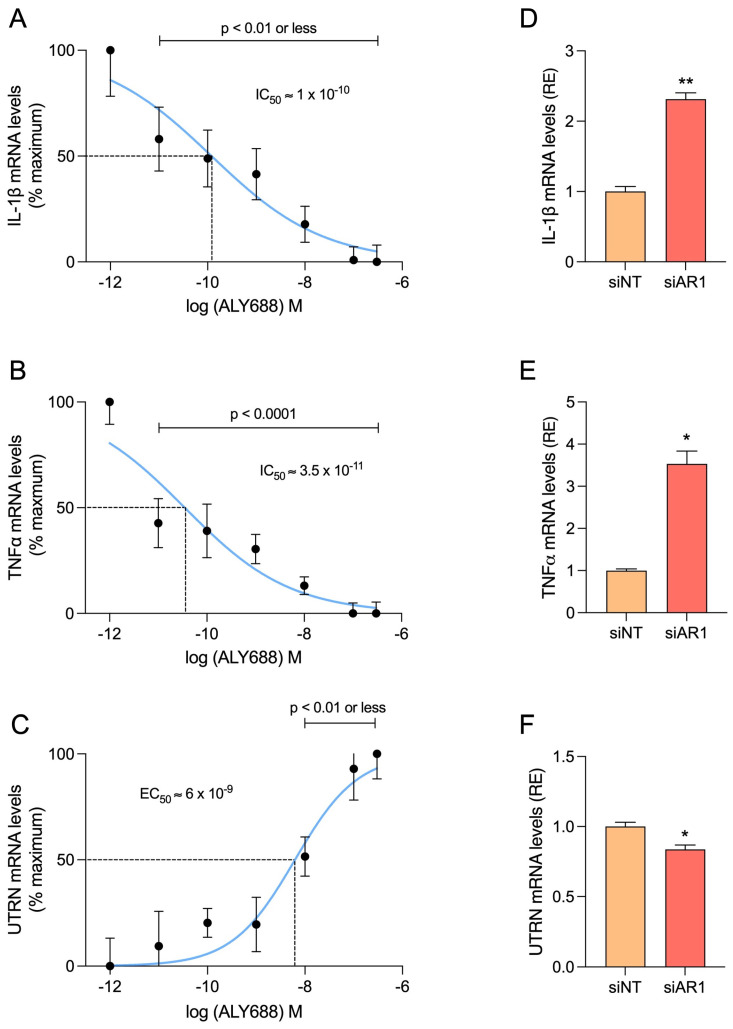
ALY688 recapitulates its anti-inflammatory and pro-UTRN effects in human DMD myotubes, via its action on AdipoR1. (**A**–**C**) Dose-response curves illustrating the effects of ALY688 on *IL-1β*, *TNFα* and *UTRN* mRNA levels in primary cultures of myotubes obtained from DMD patients. Cells were treated or not with several concentrations of ALY688 (from 10 pM to 300 nM) for 24 h, while being challenged with an inflammatory cocktail (human recombinant TNFα/INFγ, each at 15 ng/mL). mRNA levels were normalised to human *TBP*. Data were then presented as % of the maximal levels obtained either without (**A**,**B**) or with 300 nM ALY688 (**C**). (**D**–**F**) In some experiments, cells were first transfected (24 h) with siRNA against AdipoR1 (50 nM) or a negative [non-targeting, siNT (50 nM)] control and then treated with ALY688 (100 nM) combined to inflammation (TNFα/IFNγ) for an additional 24 h. After normalisation, mRNA levels were presented as relative expression (RE) to siNT conditions (**D**–**F**). Data are means ± SEM for 4 cultures, each obtained from a different donor (i.e., 4 DMD subjects). Statistical analysis was performed using repeated measures of ANOVA followed by Dunnett’s test (**A**–**C**) or a two-tailed paired Student’s *t*-test (**D**–**F**). * *p* < 0.05, ** *p* < 0.01 vs. siNT.

**Figure 9 cells-12-02101-f009:**
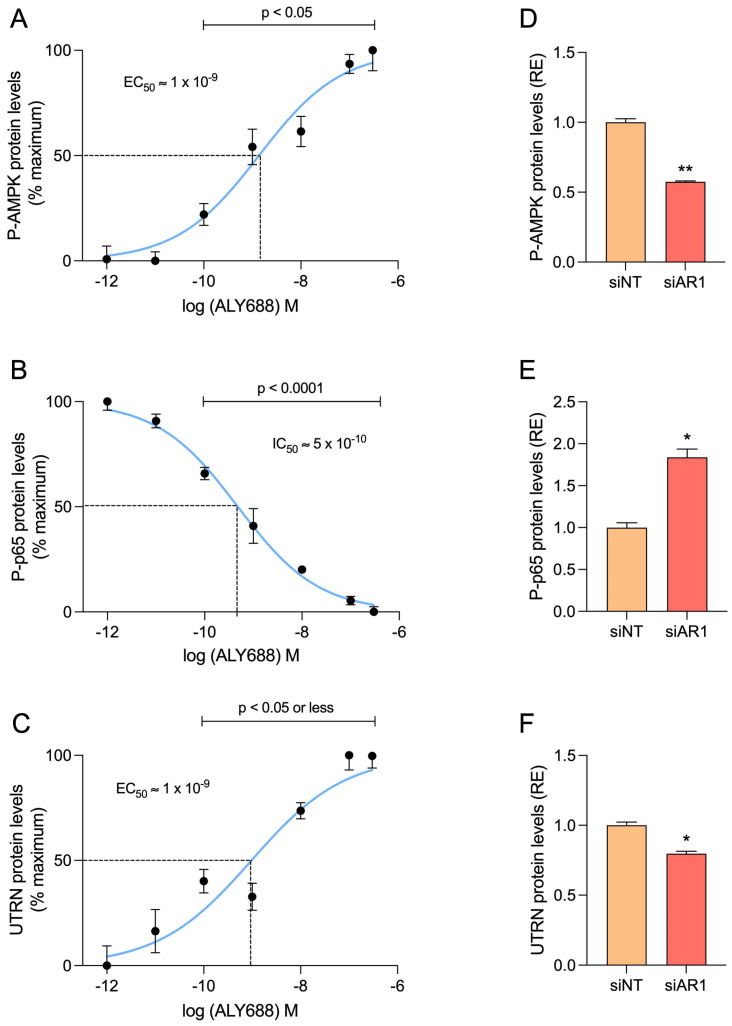
ALY688 treatment recapitulates its effects on key effectors of the AMPK signalling in human DMD myotubes, via its action on AdipoR1. (**A**–**C**) Dose-response curves illustrating the effects of ALY688 on AMPK and NF-κB activity (P-p65 subunit) and UTRN protein levels in primary cultures of myotubes obtained from DMD patients. Cells were treated or not with several concentrations of ALY688 (from 10 pM to 300 nM) for 24 h, while being challenged with an inflammatory cocktail (human recombinant TNFα/INFγ, each at 15 ng/mL). Levels of each protein were measured by ELISAs and then presented as % of the maximum achieved either without (**B**) or with 300 nM ALY688 (**A**–**C**). (**D**–**F**) In some experiments, cells were first transfected (24 h) with siRNA against AdipoR1 (50 nM) or a negative [non-targeting, siNT (50 nM)] control and then treated with ALY688 (100 nM) combined to inflammation (TNFα/IFNγ) for an additional 24 h. For each protein, levels were presented as relative expression (RE) compared with siNT conditions. Data are means ± SEM for 4 cultures, each obtained from a different donor (i.e., 4 DMD subjects). Statistical analysis was performed using repeated measures of ANOVA followed by Dunnett’s test or using a two-tailed paired Student’s *t*-test. * *p* < 0.05, ** *p* < 0.01 vs. siNT.

**Figure 10 cells-12-02101-f010:**
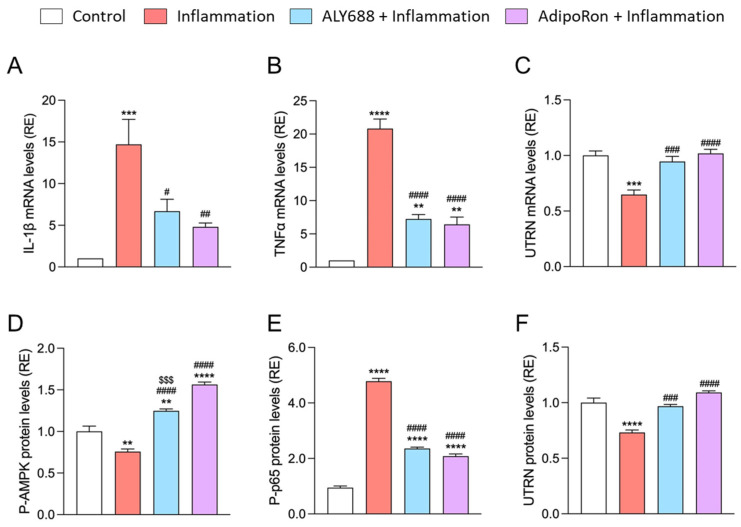
Comparison of two adiponectin receptor agonists (ALY688 and AdipoRon) in challenged myotubes from DMD patients. Myotubes were either left untreated or treated with AdipoRon (25 µM) or ALY688 (100 nM) for 24 h, while being challenged or not with an inflammatory cocktail [TNFα (15 ng/mL) + (IFNγ) (15 ng/mL)]. (**A**–**C**) mRNA levels of pro-inflammatory genes (*IL-1β* and *TNFα*) and *UTRN*. mRNA levels were normalised to human *TBP* and the subsequent ratios were presented as relative expression (RE) compared with the control condition (i.e., no compound, no inflammation). (**D**–**F**) Levels of AMPK and NF-κB activity (P-p65 subunit), and UTRN protein, were quantified by ELISAs and presented as relative expression (RE) compared with control. Data are means ± SEM for 4 cultures, each obtained from a different donor (i.e., 4 DMD subjects). Statistical analysis was performed using a one-way ANOVA followed by Tukey’s test. ** *p* < 0.01, *** *p* < 0.001, **** *p* < 0.0001 vs. Control. ^#^
*p* < 0.05, ^##^
*p* < 0.01, ^###^
*p* < 0.001, ^####^
*p* < 0.0001 vs. Inflammation. ^$$$^
*p* < 0.001 vs. AdipoRon.

**Figure 11 cells-12-02101-f011:**
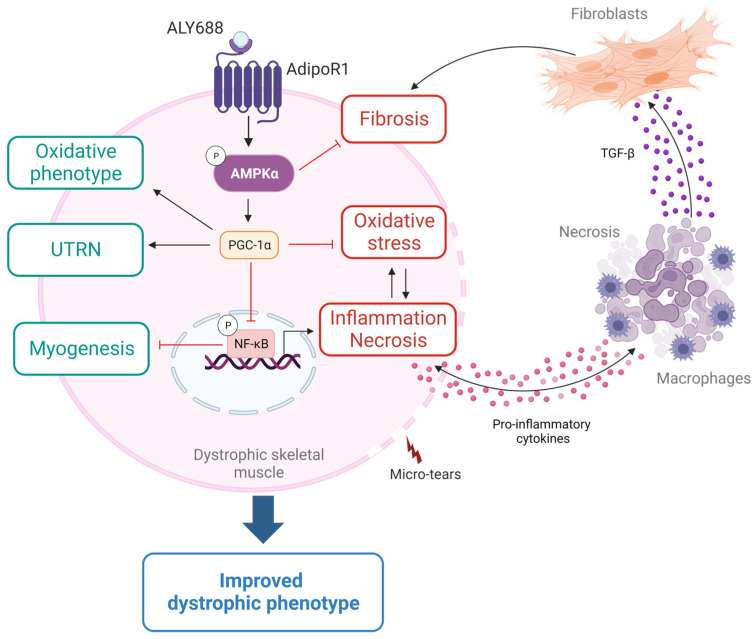
Proposed model for the effects of ALY688 in DMD. This figure summarises the effects and the mechanism of action of ALY688 on the dystrophic skeletal muscle, which is characterised by micro-tears in the sarcolemma due to lack of dystrophin protein. Briefly, binding of ALY688 to AdipoR1 will activate AMPK-PGC-1α pathway. Then, PGC-1α represses NF-κB activity resulting in a reduction in inflammation and necrosis, as well as in an improved myogenic program. In addition, the activation of the AMPK-PGC-1α axis will help mediate several effects of ALY688. First, increased muscle oxidative capacity and function. Second, increased expression and production of utrophin (UTRN) and reduced oxidative stress, which would protect the dystrophic muscle. Third, marked decrease in TGF-β levels and signalling pathways, either directly or indirectly by reducing inflammation, and subsequently blunted muscle fibrosis. These beneficial and protective properties will thus lead to an improved dystrophic phenotype. All these effects have been demonstrated on skeletal muscle from mdx mice and/or confirmed in human DMD myotubes. Pointed head black arrows indicate activation or induction, while blunt head red arrows indicate inhibition. Boxes with processes in green represent net beneficial effects of ALY688, while boxes with processes in red represent deleterious factors inhibited by ALY688. Created with BioRender.com.

## Data Availability

The raw data supporting the conclusions of this article will be made available by the authors, without undue reservation.

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
