# Peer review of "The Adiponectin Receptor Agonist, ALY688: A Promising Therapeutic for Fibrosis in the Dystrophic Muscle"

_cells, 2023, doi:10.3390/cells12162101_

Round 1
Reviewer 1 Report (Previous Reviewer 2)
The authors responded to my concerns adequately.
Author Response
We thank the reviewer for the time and effort they devoted to reviewing our manuscript, and for all the valuable comments and suggestions that helped us improve the quality of our manuscript.
Reviewer 2 Report (Previous Reviewer 1)
Line 261-262: “The wire test gives an indication on muscle force…”.
Reviewer comment: As indicated before, the wire test does not evaluate muscle force. You can argue that it evaluates motor function, muscle fatigue and/or behavior. Please reword the statement.
Line 356-357: “Myonecrosis was significantly reduced in both treated groups (~-80%) compared to regular mdx (Figure 4A, B).”
Reviewer comment: fig. 4 panel B – please note that myonecrosis was not decreased by 80% compared to both groups. Please correct the statement.
Line 384-385: “There were almost no RFs in Q sections of mdx mice, while mdx-T3 and -T15 mice displayed 2.5 to 3 times more immuno-positive fibres, respectively (Figure 5A, B).”
Reviewer comment: fig. 5 panel B – please note that dystrophin RFs in Q sections are at least 4x greater in treated groups vs mdx. Please correct the statement.
Author Response
We sincerely thank the reviewer for pointing out the small errors remaining in the manuscript. We have now corrected them and the changes are highlighted in red:
- (p6 Line 249): muscle force (wire test) is now replaced by muscle fatigue and coordination. Statement has been corrected
- (p9 Line 319): Myonecrosis was significantly reduced in both treated groups (~-60% instead of -80%). Statement has been corrected
- (p10 Lines 346-347): mdx-T3 and -T15 mice displayed 4 to 5 times (instead of 2.5 to 3 times) more immuno-positive fibres, respectively. Statement has been corrected
We thank the reviewer for the time and effort they devoted to reviewing our manuscript, and for all the valuable comments and suggestions that helped us improve the quality of our manuscript.
This manuscript is a resubmission of an earlier submission. The following is a list of the peer review reports and author responses from that submission.
Round 1
Reviewer 1 Report
Major issues:
1) One of the major issues was the use of different muscles for different assessments throughout the manuscript without a plausible explanation for its need. It has been shown by different studies that the outcomes evaluated in the current study (IF, IHC, mRNA, protein levels) can be measured from the same muscles (serial muscle sections can be cut and used for different assays). Thus, the authors should make it clear throughout the entire manuscript that muscle X was used for outcome Y.
2) Need to add catalog number for ELISA kits, and show a previous reference using the same assay on mdx samples. For instance, what previous work has used an ELISA to evaluate UTR levels in muscles from mdx mice?
3) Is there an explanation for the following statistical comparison: “For the grip test, comparison between mdx and mdx-T15 mice was carried out by an unpaired two-tailed t-test, $$ P < 0.01 vs mdx mice.” As stated by the authors, “Statistical analysis was performed using one-way ANOVA followed by Tukey's test” as this should be the correct statistical test for fig. 1, unless the authors can provide an explanation for choosing a t-test for only one comparison.
4) Need to add the number of mice/samples used per outcome or bar graph in figures. Also, while in the methods the authors stated that 10 mice were used per group, some of the outcomes were only evaluated in 6 mice per group (e.g., fig 4, 5 and 6). The authors need to explain why only data from 6 mice are shown.
5) TNFa is a well-known marker of inflammation in dystrophic muscles, and the results clearly show that TNFa % area was not different between groups, which shows that ALY688 has a limited effect on muscle inflammation. This limits the implications of the study, and the authors should make it clear throughout the manuscript that ALY688 has somewhat limited anti-inflammatory effects on mdx mice. The same applies for oxidative stress – the data from HNE was not significantly different, so the interpretation of the results should be clear that ALY688 does not decrease overall oxidative stress in dystrophic muscles.
6) Since the authors stated “Because inflammation and cell damage may subsequently lead to muscle regeneration, we hypothesised that ALY688 could promote a muscle healing process.”, they should have included data evaluating the areas of muscle damage/necrosis as well as %CNFs per total muscle section. This will greatly improve the quality of the manuscript as is easily achievable since the authors already have images for analysis. Also, the reference used to describe the relationship between RF and muscle regeneration (ref # 24) clearly used a different method to quantify the number of RFs compared to the current manuscript – they evaluated total number of RFs, number of clusters, and max number of RF in a cluster. Additionally, authors from ref #24 also evaluated %CNFs in their study.
7) Gene expression data from section 3.3, on Mrf4, Myh1, and Myh7, do not support that the treatment increases late differentiation or muscle regeneration.
8) The summary from section 3.3 “Taken together, these results indicate that ALY688 might induce some regeneration capacity in dystrophic muscle.” clearly show that more work is needed to not only improve this section but also the quality of the manuscript. For instance, evaluate areas of degeneration/necrosis and %CNFs, as well as adding IF and quantification of other markers of myogenesis such as Pax7 and myogenin.
9) Discussion written in a speculative way; should have added comparisons of findings with other studies evaluating similar outcomes to better show the promising effects of the drug for DMD. Also, needed to discuss results that were not significantly different.
Minor issues:
1) The authors stated that the treadmill exhaustion test was performed “uphill treadmill exhaustion test with a downward inclination of 5° and increasing speed rate over 4 steps”. However, the methods did not show that the treadmill grade was changed, while only the speed was altered throughout the test.
2) What was the protocol used for IHC?
3) Need to show body weight data – especially if there is a difference between WT and other groups.
4) “The wire test gives an indication on muscle force” – no; while it might evaluate muscle fatigue resistance, it’s also a behavioral test. It does not evaluate muscle force per se.
5) In Fig. 1 – “global force” – change it to “muscle function”. Again, not all tests are evaluating muscle force, so its better to add muscle function.
6) Can the authors speculate as to why the drug does not have a dose-dependent effect in the outcomes evaluated?
7) In the text the authors stated: “Quantification of DAB staining confirmed that these parameters in mdx-treated mice were either significantly decreased compared to regular mdx (IL-1β: -54% and -33%; PRDX3: -49% and -43%; CD68: -47% and -44% in mdx-T3 and mdx-T15, respectively)”. However, IL-1β was only statistically significant different between mdx and mdx-T3.
8) Fig. 2 “Representative sections for 6 mice per group are shown.” – only four images per target are shown.
9) Figure S2 – while the experiments demonstrate that ALY688 blunts the detrimental effects of inflammation on Mrf4, Mrf7, and ERRa mRNA levels, the authors should have performed experiments quantifying the levels myogenin and fusion index in myotubes.
10) Fig. 3 - Why was IL-1B so high in WT muscles? How does it compare to other studies?
11) TGFb ELISA – did it measure total active TGFb levels or total TGFb protein levels? If the assay did not measure active TGFb levels, then recommend adding such assay to determine whether ALY688 can decrease active TGFb levels from muscle from treated mice.
Reviewer 2 Report
This is a well-designed study of a well-known group. Importantly, the effects of ALY688 were tested not only on mdx mice showing a mild phenotype, but also on human DMD myotubes. I have a few questions and recommendations:
1. Lines 90-91. Is it possible to provide a Ref. to these results?
2. Lines 196-197. These primers should be given again for the convenience of the reader.
3. In what order were the physiological tests performed? Are they all done on the same day? This should be noted in the materials and methods section, as it may affect the reproduction of results.
4. Why does the mdx-T15 group show some decrease in the parameter in the grip test? In addition, a large concentration does not affect the change in inflammation and other parameters. What could be the reason for this? It is advisable to discuss this.
5. I know that your research has been sent to the editor before, but it would be very helpful to discuss your work in the context of the doi.org/10.1101/2023.05.22.541826 preprint published a few days ago.
6. The authors briefly note that PGC-1α is involved in mitochondrial biogenesis. However, this is not disclosed in the discussion. It is known that DMD is accompanied by suppression of organelle biogenesis and their function (both oxphos and a decrease in the efficiency of calcium and potassium ion transport and resistance to the MPT pore), which has an important effect on the progression of pathology. It is also known that overactivation of PGC-1α with givinostat leads to an improvement in organelle biogenesis and function and also mitigates the development of pathology. Such a mechanism should also be discussed.